# Broadband Terahertz Detection by Laser Plasma with Balanced Optical Bias

**DOI:** 10.3390/s22197569

**Published:** 2022-10-06

**Authors:** Xu Sun, Zhi-Hui Lyu, Hai-Zhong Wu, Cong-Sen Meng, Dong-Wen Zhang, Zhi-Zhong Lu, Xiao-Wei Wang, Zeng-Xiu Zhao, Jian-Min Yuan

**Affiliations:** 1Department of Physics, National University of Defense Technology, Changsha 410073, China; 2Key Laboratory of Robotics Engineering and Intelligent Manufacturing, Southwest Petroleum University, Nanchong 637001, China; 3Graduate School of China Academy of Engineering Physics, Beijing 100193, China

**Keywords:** broadband terahertz detection, laser plasma, optical bias, terahertz-induced second harmonic

## Abstract

Using a controlled optical bias and balanced geometry, we propose a new scheme for broadband terahertz detection by laser-gas interaction without high-voltage manipulation. Compared to the conventional optical bias scheme, the common noise is reduced and the dynamic range as well as the signal-to-noise ratio are doubled. It provides a simple alternative for coherent broadband terahertz detection. The influence of optical bias on terahertz waveform is also investigated, and the evolution of the terahertz-induced second harmonic with probe delay is further revealed. This new detection scheme for broadband terahertz will boost the application of terahertz time-domain spectroscopy for its miniaturization and integrability.

## 1. Introduction

Utilizing the interaction of ultrafast lasers with gases, it is possible to obtain intense terahertz (THz) radiation with continuous spectrum from far-to-mid infrared [1,2,3,4,5]. This radiation has important implications for materials research in physics and chemistry because its photon energy couples various excitations, such as molecular vibration and rotation, phonon, and intraband transition. Air-biased coherent detection (ABCD) [6,7], which in a sense can be considered as the inverse process of THz generation, is often employed to detect such broadband THz pulses [8,9,10,11]. Compared with a photoconductive antenna [12] and electro-optic sampling [13], ABCD does not rely on specialized optical material, has no damage threshold, and can achieve a relatively smooth response over a very wide bandwidth. It converts the THz wave to the second harmonic of the laser pulse, employing the third-order optical nonlinearity of the gaseous medium [6]. The THz electric field is reconstructed by the interference between such a THz-induced second harmonic (TISH) and another harmonic acting as the local oscillation (LO) [6,7]. Early implementations of the LO were obtained by means of spectral broadening though self-phase modulation and self-steepening [6]. As the supercontinuum in air plasma requires a very high amount of laser power and cannot be independently controlled, in terms of polarization, phase, and bias intensity, an alternating-polarity bias was appended and used as the LO [7]. In such a scheme, a non-universal high-voltage source is needed and synchronization with the laser is also required. To avoid the high-voltage handling, an attempt with controlled optical bias was made, in which the LO was fully controllable [14]. Because the LO plays a key role in heterodyne detection, improved optical bias provides an effective way for the optimization of terahertz detection.

Here, we present a new scheme for THz detection employing optical bias. It combines the merits of optically biased coherent detection (OBCD) [14] and balanced detection [13]. The intensity and phase of the LO are finely controlled, and TISH interferes with two mutually conjugate LO beams followed by differential detection. The common mode noise from a laser can be effectively suppressed. Although a balanced ABCD had been proposed [15], the previous scheme relies on the third-order nonlinear tensor symmetry of gases and is susceptible to cross-phase modulation in the laser-gas interaction [16]. In our scheme, a high-voltage source and electrode pairs are unnecessary; all the optical and electrical components are concentrated behind the focus of the THz and probe beams, which makes it easy to be integrated into a stand-alone module and convenient in applications. In addition, the scheme is also suitable for the recently reported water-based coherent detection [17].

## 2. Experiment Setup

Our experimental scheme is shown in Figure 1. Two-color pump, in which the fundamental and second-order harmonics are linearly polarized along the same direction, is obtained through a co-line phase compensator [18]. It is optically chopped at half the laser repetition frequency and focused by an off-axis parabolic mirror to produce THz emission. Another two off-axis parabolic mirrors are used to collimate and refocus THz beam, respectively. The residual pump is filtered by a high-resistivity silicon wafer. The probe, polarized identically to the pump and sent from a time-delay stage, is focused through a hole in the last parabolic mirror and collinearly propagates with the THz beam. The focuses of the two beams are overlapped spatially to produce TISH. TISH will be polarized in the same direction as the residual probe, as illustrated by the blue and red arrows behind the fused silica plate in the figure. A dichroic mirror with high transmissivity for TISH and angle-dependent reflectivity for fundamental beam is designed. Perpendicularly polarized LO is generated from a β−BBO crystal by type I phase matching. The intensity and time delay of LO can be finely controlled by tilting the dichroic mirror and the followed fused silica plate. A lens is used to reduce the divergence of TISH and LO. Time synchronization of the two cross-polarization pulses is achieved utilizing the difference in group velocities in the fast and slow axis directions in an α−BBO crystal. The birefringence in the α−BBO is exploited to temporally advance the 2ω bias envelope in the experiment. Then, TISH and LO are converted to left- and right-handed circularly polarized beam, respectively, by a quarter-wave plate. Such two beams with conjugate polarization pass through a Wollaston prism and are both divided into two components with perpendicular polarizations. The components in each polarization direction interfere and are detected differentially by two photomultiplier tubes (PMTs). To eliminate the effects of the residual probe and external light, the PMTs are fitted with 400 nm bandpass filters.

According to the four-wave mixing mechanism of TISH generation [6,7], the TISH field $E_{\mathrm{TISH}}$ in the scheme is proportional to the THz field $E_{\mathrm{THz}}$. It can be expressed as:(1)$$E_{\mathrm{TISH}}\propto \chi^{\left(3\right)}E_{\omega }^{2}E_{\mathrm{THz}},$$
where $E_{\omega }$ and $\chi^{\left(3\right)}$ are the probe field and the third-order susceptibility of the gas, respectively. If LO is introduced, the intensity of second-harmonic $I_{2\omega }$ after interference between TISH and LO is:(2)$$I_{2\omega }\propto I_{\mathrm{LO}}+I_{\mathrm{TISH}}+E_{\mathrm{TISH}}E_{\mathrm{LO}}\cos\left(\Delta \varphi \right),$$
where Δφ denotes the phase between TISH and LO. In ABCD detection, an alternating–polarity bias is employed [7]. The amplitude of THz field is obtained by the lock-in technology in response to the third term of Equation (Equation 2). While in OBCD detection, the second term is suppressed in the case that LO is much stronger than TISH, and the first term is eliminated by the lock-in detection [14].

In our scheme, TISH and LO, whose polarization directions are perpendicular to each other, are converted into circularly polarized beam with opposite helicity by the quarter-wave plate. The Jones matrix for the synthetic vector of the two beams is:(3)$$E_{\mathrm{TISH}}\left[\begin{array}{c} \frac{\sqrt{2}}{2}\\ \frac{\sqrt{2}}{2}j \end{array}\right]+E_{\mathrm{LO}}\left[\begin{array}{c} \frac{\sqrt{2}}{2}\\ -\frac{\sqrt{2}}{2}j \end{array}\right]e^{j\Delta \varphi }.$$

The intensities of two perpendicular components split by the Wollaston prism can be denoted as:(4)$$I_{2\omega }^{\left(p\right)}\propto I_{\mathrm{LO}}+I_{\mathrm{TISH}}+E_{\mathrm{TISH}}E_{\mathrm{LO}}\cos\left(\Delta \varphi \right),$$
(5)$$I_{2\omega }^{\left(s\right)}\propto I_{\mathrm{LO}}+I_{\mathrm{TISH}}-E_{\mathrm{TISH}}E_{\mathrm{LO}}\cos\left(\Delta \varphi \right),$$
where the superscript *p* and *s* correspond to the two polarization directions. If the output of a single PMT is sent directly to the lock-in amplifier as the input, the detection setup can be considered as an equivalent of OBCD. To suppress the common mode noise in detection, we use a balanced detection method similar to the one used in electro-optical sampling. The differential signal between the outputs of two PMTs is employed as the input to the lock-in amplifier.

The experiment was performed in ambient air with a Ti:sapphire-amplified laser system which delivered 800 nm, 25 fs, 1 mJ pulses at a 1 kHz repetition rate. A total 0.8 mJ of pulse energy was used as the pump for THz generation, and 0.1 mJ of energy was used as the probe for THz detection. The thicknesses of the dichroic mirror, fused silica plate, α−BBO (z−cut), β−BBO, and the substrate of β−BBO were 1 mm, 2 mm, 1 mm, 10 µm, and 1 mm, respectively.

In our scheme, the β−BBO crystal providing LO is placed behind the probe focus. As the phase distortions of TISH and residual probe arising from propagation in plasma are inevitable, the use of the residual probe to obtain LO can make the wave fronts of TISH and LO similar, thus diminishing the influence of gas ionization on the interference between the two beams. We had tried to place the optical elements which generate and control LO, including dichroic mirror, fused quartz plate, β−BBO, and α−BBO, between the parabolic mirror and the lens. The second-harmonic intensity from the interference terms in Equations (Equation 4) and (Equation 5) is about 6 times weaker than the optically biased LO intensity.

## 3. Results and Discussion

Figure 2a shows the THz waveforms obtained with such a balanced detection in the experiment, as well as those obtained with each single detector. The waveform of TISH is also given, which was obtained with a single detector after removing the β−BBO. The integration time of the lock-in amplifier is set to be 100 ms in the experiment. The two waveforms obtained with each single detector correspond to the last two terms in Equations (Equation 4) and (Equation 5), respectively. Although they are superimposed with TISH, the small amplitude of TISH (about 45 times weaker than that of the OBCD signal) makes the two waveforms appear to have the same amplitude and opposite polarity. The balanced THz waveform, equal to the subtraction of the previous two waveforms, acquires a two-fold increase in amplitude. Figure 2b shows the spectra from the windowed Fourier transform of the THz waveforms. The spectra indicate a broad bandwidth covering from 0.5 to 30 THz. The background noise obtained with the balanced detection is almost the same as that obtained with the single detector, which implies that the noise from the LO is well filtered out by the lock-in technique. Because the differential detection doubles the signal amplitude, a gain of 6 dB has been achieved in the spectrum of the balanced detection. To reveal the enhancement of the signal-to-noise ratio (SNR), 36 measurements of the THz waveform were made. The stability of the THz spectra after the Fourier transform was compared. As shown in Figure 2b, the shaded errorbars indicate the jitter ranges of the THz spectra. The jitters are frequency-dependent and strongly modulated by the water vapor absorption. In comparison, the spectral jitter obtained with the balanced detection is much smaller. Taking half the jitter widths as the noise amplitudes [19], the SNR curves are shown as the dashed lines in Figure 2b, which indicate that the balanced detection performs much better than that of the single detector. Overall, the balanced detection provides a noise suppression capability, with which the SNR is improved to more than twice.

In the balanced detection, the signal obtained from a lock-in amplifier corresponds to the difference between Equations (Equation 4) and (Equation 5), whose value is affected by the relative phase Δφ between TISH and the LO. We used a pair of optical wedges instead of the fused quartz plate in Figure 1. The relative phase could be precisely controlled by finely overlapping the optical wedge pair. Figure 3 shows the THz waveforms for different values of Δφ. Each horizontal line in the contour plot represents the THz waveform at a certain Δφ. The color depths of red and blue indicate the strength of the positive and negative THz fields, respectively. Similar to the experimental result of Li et al. [14], the THz waveform varies with Δφ with a period of 2π. There is no certain Δφ such that the waveform amplitude converges to zero. It means that the phase of TISH is not a constant but varies simultaneously with the amplitude as the probe delay changes. The same may also be true for the TISH spectra.

To reveal such information about TISH, we measured the TISH spectra with a monochromator. The result is shown in Figure 4. As the measurements were time-consuming, only the spectra corresponding to the probe delay in the range from −0.2 to 0.2 ps were measured. The range covers the transition of the THz field from the positive to negative maximum in the time domain. It shows that the intensity and spectrum distribution of TISH both varies with the probe delay, but the intensity of TISH does not converge to zero simultaneously with the THz field. The signatures support our previous speculation that the phase of TISH varies with the probe delay. The TISH spectra at three probe delays are shown in Figure 4c, corresponding to the probe delays at which the TISH intensity (solid blue) reaches its maximum and the THz field reaches the positive (dashed orange) and negative (dashed purple) maxima, respectively. Compared with the LO, TISH has a much broader spectrum. The modulations on these spectral profiles may be caused mainly by the spectral changes in the probe due to self-phase modulation.

At the 0 ps instant in Figure 4a, the TISH intensity reaches its maximum. The redshift of the spectral peak is accompanied by a broadband extension toward the long wavelength direction. It reveals that the $2\omega =\omega +\omega -\Omega_{\mathrm{THz}}$ process dominates the TISH generation, as suggested by Lu et al. [14,20,21]. Comparing the spectra offset of the LO and TISH, it can be inferred that the highest THz frequency in the four-wave mixing is at least 40 THz or more, much higher than that measured in the experiment. It is notable that the THz waveform obtained by such heterodyne detection is minimally affected around 0 ps, although the intensity and spectrum of TISH vary strongly. Considering that the spectral distributions of TISH and the LO have a good overlap, it can be inferred that the chirps of TISH and the LO are different at this probe delay, which leads to a weak interference between the two beams. Further, the weak interference results in a weak response to the high-frequency components of the THz emission. To overcome this problem in gas-based THz detection, the interference between the LO and TISH needs to be enhanced. It can be achieved by controlling the spectrum and chirp of the LO. Attempts have been made to achieve ultra-broadband spectrum detection by using the shortest probe as possible [22,23]. In fact, optimizing the spectrum and chirp of the LO is the most essential approach.

Due to the variation of the phase of TISH with the probe delay, the THz waveforms obtained from the interference of TISH and the LO are related to the phase of LO. A contrast of the THz waveforms corresponding to different LO phases is given in Figure 5a. Each curve represents the THz waveform at a certain LO phase. The light purple background indicates the envelope of these THz waveforms. It suggests that this type of heterodyne detection used in the broadband THz time-domain spectrometer is susceptible to the delay of the THz waveform inevitably introduced by the inserted samples, unlike electro-optical sampling and a photoconductive antenna.

The range of the relative variations in the THz spectra resulting from the the change in the LO phase is given in Figure 5b. A certain spectrum is chosen as the reference to obtain the scaling factor for the THz spectra. We measured the THz spectra by scanning the controllable phase and normalized to the reference spectrum to obtain the frequency-dependent scaling factor. The green shading in Figure 5b indicates the range of the scale factor. The wide range distribution in the high-frequency region (>25 THz) is caused by the low signal-to-noise ratio. The scale factor in the low-frequency region (<25 THz) is distributed between 0.9 and 1.5. It shows that the THz spectrum is less variable despite the more dramatic changes in the time-domain waveform. Surely, the enhancement of the THz spectral intensity by the optimal LO phase is also meaningful for practical applications.

## 4. Conclusions

We have demonstrated a gas-based THz detection scheme using a controlled optical bias and balanced geometry. Employing balanced detection, the dynamic range and signal-to-noise ratio of THz spectra are enhanced by a factor of about two in comparison with the conventional optically biased detection. In addition, the scheme is compact without a high-voltage source or electrodes. The size can be further reduced if avalanche phototubes are used instead of photomultiplier tubes [24], which makes it possible to integrate the whole detection module into a bundled assembly. A further study of TISH shows that the intensity, phase, and spectrum of TISH all change with the probe delay, which makes this heterodyne detection a relative one. Optimizing the LO spectra and chirp to improve the interference between the LO and TISH, a wider detection bandwidth surely can be achieved.

## Figures and Tables

**Figure 1 sensors-22-07569-f001:**
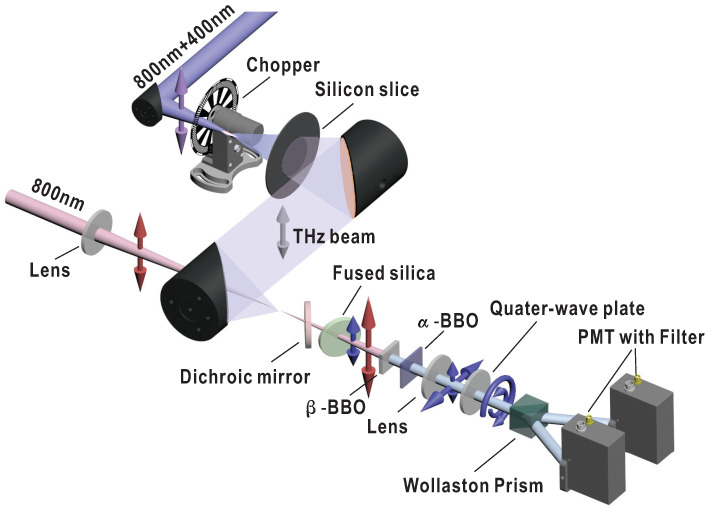
Experimental scheme using controlled optical bias and balanced geometry. BBO: barium borate. PMT: photomultiplier tube. Dichroic mirror, fused silica, β−BBO, and α−BBO crystals are used to provide controllable LO, while quarter-wave plate, Wollaston prism, two PMTs are used for balanced detection. The polarizations of 400 nm and 800 nm pulses are indicated by the blue and red arrows, respectively. TISH is divided into two parts, which interfere with two mutually conjugate LOs, respectively. The differential signal between the outputs of two PMTs is employed as the input to a lock-in amplifier.

**Figure 2 sensors-22-07569-f002:**
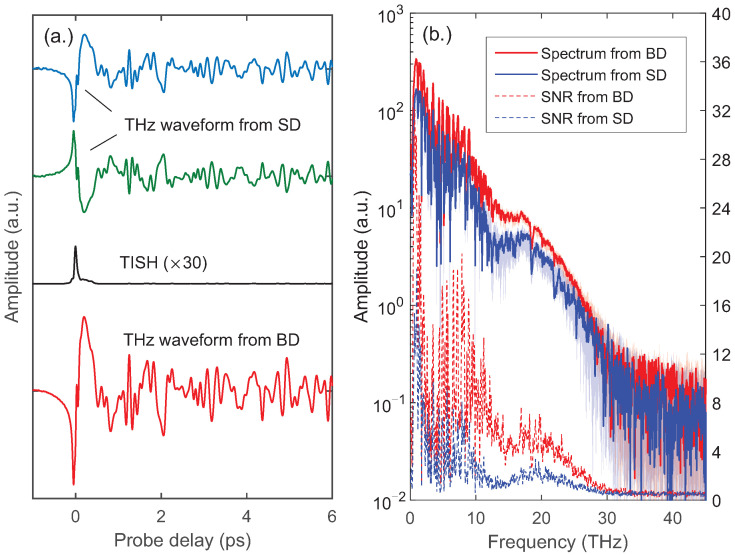
(**a**) THz waveform obtained with balanced detection (BD), compared with the waveform of TISH and the THz waveforms detected with each single detector (SD). The waveform offsets are shifted for clarity. TISH was detected with a single detector after removing the β−BBO in the scheme. (**b**) THz spectra (solid curves, left axis) obtained from windowed Fourier transform of THz waveforms. The shaded errorbars indicate the jitter ranges of the THz spectra in 36 measurements. The SNR curves are shown as the dashed lines, taking half the jitter widths as noise amplitudes (right axis).

**Figure 3 sensors-22-07569-f003:**
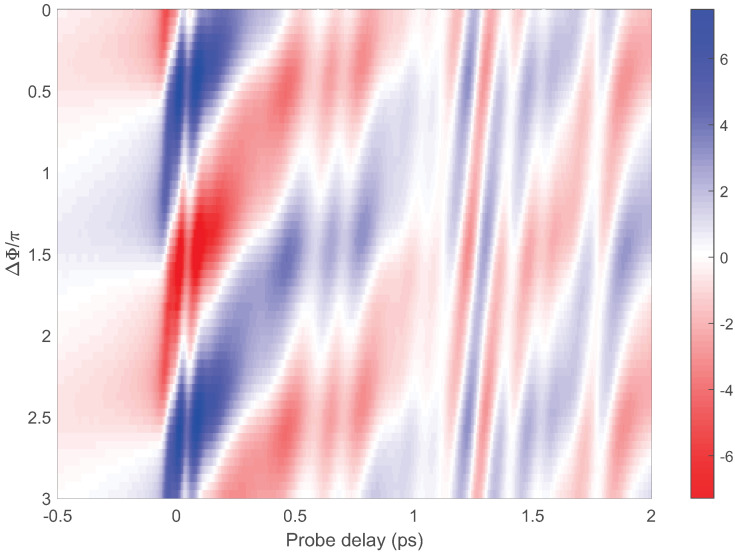
Time-domain THz signals from balanced detection with different values of Δφ. The THz waveform corresponding to each Δφ is shown as a horizontal line on the contour plot, where the color depths of red and blue indicate the strength of the positive and negative fields, respectively.

**Figure 4 sensors-22-07569-f004:**
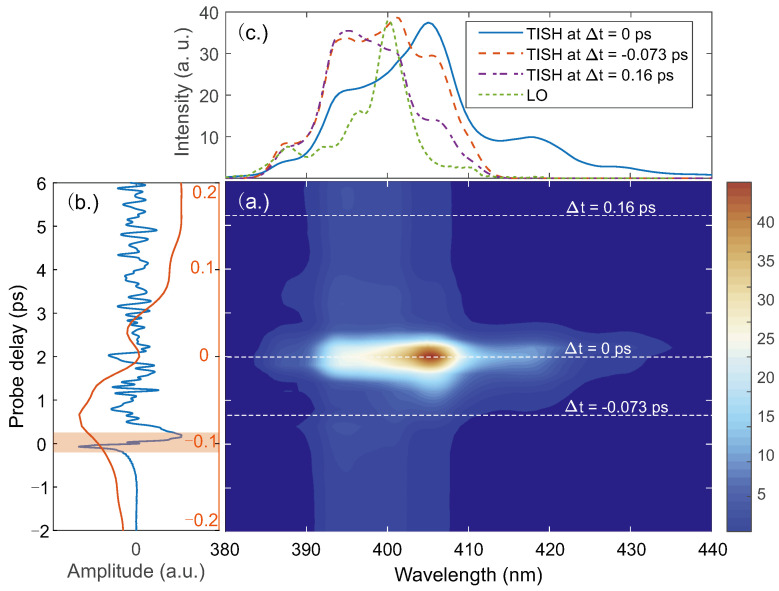
(**a**) The TISH spectra measured with a monochromator corresponding to the probe delay in the range from −0.2 to 0.2 ps. The dashed white lines mark the three probe delays at which TISH is measured at 0 ps, −0.073 ps, and 0.16 ps, corresponding to the probe delays when TISH intensity peaks and the terahertz electric field reaches the positive and the negative maxima, respectively. (**b**) The orange THz waveform recorded within the same range of the probe delay in (**a**). The blue THz waveform is measured within a longer range of probe delay. The shaded region shows the time slot taken by the orange THz waveform. (**c**) The TISH spectra at three probe delays in (**a**). For comparison, the spectrum of LO (dashed green) was measured with attenuated intensity.

**Figure 5 sensors-22-07569-f005:**
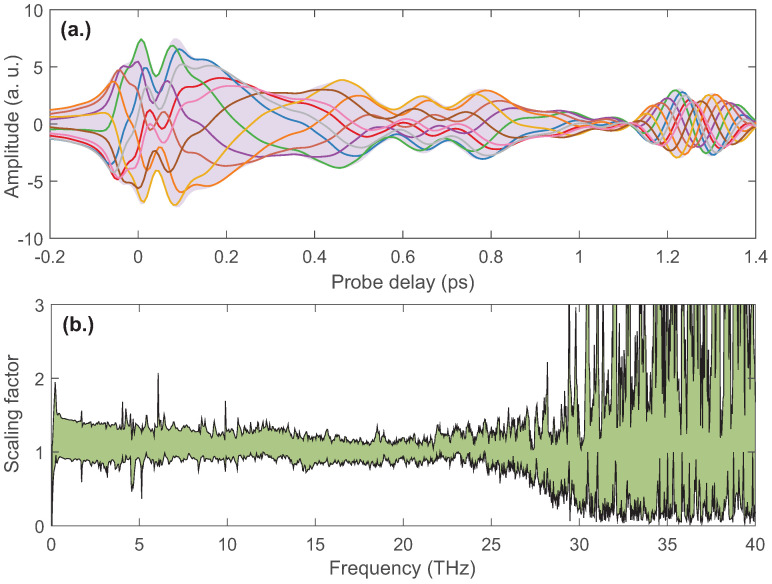
(**a**) The THz waveforms corresponding to different LO phases. Each curve represents the THz waveform at a certain LO phase. The light purple background indicates the envelope of these THz waveforms. (**b**) THz spectral variation with the LO phase. A certain spectrum is chosen as the reference to obtain the intensity scaling factor for the THz spectra. The green shading in the figure indicates the range of the scale factor.

## Data Availability

The data that support the plots within this paper are available from the corresponding author upon reasonable request.

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
