# Peer review of "Broadband Terahertz Detection by Laser Plasma with Balanced Optical Bias"

_sensors, 2022, doi:10.3390/s22197569_

Round 1

Reviewer 1 Report

Dear Authors,

The intention of your work is to attempt to improve the performance of THz air-biased coherent detection i.e., THz ABCD, by applying optically biased balanced detection. However, there are quite a lot of problems in their manuscript, including some conceptual misunderstandings, as listed below. I won’t suggest the publication of the manuscript on Sensors based on these comments.

1.               First, the authors claim that the dynamic range and signal-to-noise ratio of the THz ABCD is significantly improved using their method. However, claiming an improvement on the SNR needs a direct comparison between the SNR of this method and those previous approaches, which is absent in this manuscript. In fact, the SNR obtained from the “optically biased” approach in this manuscript seemingly shows no improvement over that of the “electrically biased” approach, which can be much higher than what you have obtained in this work.

2.               Both the THz waveform and the spectrum shown in Fig.2 are heavily affected by absorption of water vapor and look very blurring. Besides, the SNR in Fig.2 (b) is indistinct partially due to the water vapor absorption lines.

3.               Authors claim in the conclusion that “Employing balanced detection, the dynamic range and signal-to-noise ratio of THz spectra are enhanced by about a factor of two.” However, it has already been demonstrated that the SNR of the balanced THz ABCD based on an “electrically biased” scheme can be improved by a factor of two (Appl. Phys. Lett. 98, 151111 (2011)). The improvement factor on SNR in this work using optical bias is essentially the same as the previous “electrically biased” approach, implying there is no significant improvement in the SNR of THz ABCD at least for now. And there is no credible evidence shown in this manuscript that they can further improve the SNR in comparison with the previous work.

4.               The concept of “optical bias” claimed by authors in their manuscripts is controversial. The so-called optical bias (external second harmonic) is generated after the interaction of the THz wave and probe beam in their experiment (i.e., the optical second-harmonic doesn’t actually bias on the detection plasma), which means the optical bias is not directly applied to the detection process, leading to the misunderstanding of the detection mechanism. In fact, the two second-harmonic signals are just linearly added together (linear interference).

5.               The use of the externally induced local oscillator (LO) will couple the fluctuation of the amplified laser into the detected THz signal, which may cause a larger noise than the case of electrical bias since the ripple of a commercial high voltage power supply is far less than the fluctuation of an amplified laser. Therefore, I think the advantages of the “optically biased” scheme over the “electrically biased” scheme claimed in this manuscript are debatable.

6.               Authors stated in the text (page 6, the last sentence of the second paragraph) “It suggests that this type of heterodyne detection is a relative detection that cannot obtain the real shape of the THz wave, unlike electro-optical sampling and photoconductive antenna”, implying that EO sampling or photoconductive antenna switching can detect the real THz waveforms. I would say, the authors’ basic understanding of the detection of THz waves is problematic. Even if one uses “EO sampling or “photoconductive switching” to detect THz waves, the detected THz waveforms will be deformed (different from the original THz waveforms) due to the limited detection bandwidth, and the dispersion of the EO crystals or non-uniform transfer function of the entire THz system.

Sincerely yours,

Sensors’ Reviewer

Author Response

Thanks.

Reviewer 2 Report

This article presents a novel approach to perform coherent balanced terahertz detection using the interaction of an emitted terahertz field with an optical LO through the third order non-linearity of a gaseous plasma excited by pump. The article also analyzes the dependencies of the detection technique on the temporal delay between emitted terahertz field and the optical LO. The results are very interesting, and they are also well presented. The degree of technical sophistication is quite high.

I recommend acceptance after the following minor corrections.

1. In line 7, the word ‘spectroscoy’ should be replaced by‘spectroscopy’.

2. In the sentence written within lines 21-22 some articles are missing. I think the sentence should read: “It converts the THz wave to the second harmonic of the laser pulse employing the third-order non-linearity”.

3. The beginning of the sentence within lines 24-25 doesn’t make much sense. I recommend that the authors change it to something like: “Early implementations of the LO were obtained by means of…”.

4. In line 26, I recommend that the authors specify that the threshold requirement is about an optical threshold, i.e. that the a very high amount of laser power is required for that technique to work.

5. In line 58, I recommend substituting the word ‘repress’ by ‘reduce’.

6. In lines 82-83, spaces are missing between numerical values and units. Also, 10 um should be replaced by 10 µm.

7. In line 89, I think the authors meant ‘dichroic mirror’ and not ‘bichromatic mirror’.

8. In line 91, the article ‘the’ before ‘optical biased LO intensity’ is missing.

9. In line 111, the word ‘widthes’ should be replaced by the word ‘widths’.

10. In line 155, the verb ‘varies’ should be replaced by the noun ‘variations’.

I also recommend that the authors explain in more detail a few technical points to improve even more the quality of the article:

1. How was the difference signal between the two PMTs obtained? What electronic device did they use? A 180° hybrid coupler? What bandwidth it had?

2. The authors should explain in more detail how the time-synchronization between the two cross-polarization pulses is achieved utilizing the difference of group velocities in the fast and slow axis directions of the BBO crystal. Did they change of the angle incidence? Did they rotate the crystal? What would happen if this synchronization is not done?

3. The authors should explain why they used the jitter of the signal as the noise floor for the SNR calculation. Normally, the noise floor is the signal level at which the signal cannot be distinguished anymore. Why didn’t they use this usual definition for the noise?

4. The authors should specify the integration time they used to obtain the plots in Figure 2.

5. The time-domain signals plot in Fig. 4(b) do not seem to match. The time-domain signal with higher temporal resolution has a lot of noise after -0.1 ps while the time-domain signal exhibiting a longer delay does not seem to exhibit the same noise. Also, the slopes seem to be different.

6. The authors should specify how they chose the spectrum that they used as reference to obtain the scaling factor in Fig. 5(b).

7. The authors should be aware that although their system does not require an electrical bias to operate, it is not compact compared to other existing systems such as the ones implemented using photoconductive antennas. In particular, their system requires a lot of free-space optical components that are not required by photoconductive systems. Therefore, I suggest that the authors modify their statements about this in the Abstract and in the Conclusion.

Author Response

Thanks.

Reviewer 3 Report

In this work, the authors proposed a new scheme for broadband terahertz detection by laser-gas interaction without high voltage manipulation, by using controlled optical bias and balanced geometry. The main advantages of their proposed scheme are that the common noise is reduced, and at the same time the dynamic range and the signal-to-noise ratio are doubled, as compared with the conventional optical bias scheme. So, this work is practically helpful for miniaturization and integration of terahertz time-domain spectroscopy. I would like to recommend the work for publication, provided that the following concerns are addressed:

It is well-known that refractive-index sensors based on surface plasmon resonances in metal nanostructures are promising in label-free biomedical detection, and many works has been published, such as Nanotechnology 32(46), 465202 (2021) and J. Lightwave Technol. 39(2), 562-565 (2021). I wonder the new scheme proposed in this manuscript whether or not has potential in label-free biomedical detection? And what are the advantages of the proposed scheme, as compared with well-known refractive-index sensors?

Round 2

Reviewer 1 Report

Please check the attached pdf.
